# Synthetic Lethality Screening Highlights Colorectal Cancer Vulnerability to Concomitant Blockade of NEDD8 and EGFR Pathways

**DOI:** 10.3390/cancers13153805

**Published:** 2021-07-28

**Authors:** Federica Invrea, Simona Punzi, Consalvo Petti, Rosalba Minelli, Michael D. Peoples, Christopher A. Bristow, Valentina Vurchio, Alessia Corrado, Alberto Bragoni, Caterina Marchiò, Andrea Bertotti, Livio Trusolino, Alberto Bardelli, Claudio Isella, Alessandro Carugo, Giulio F. Draetta, Enzo Medico

**Affiliations:** 1Candiolo Cancer Institute, FPO-IRCCS, 10060 Candiolo, Italy; federica.invrea@ircc.it (F.I.); Punzi.Simona@hsr.it (S.P.); consalvo.petti@ircc.it (C.P.); valentina.vurchio@ircc.it (V.V.); alberto.bragoni@ircc.it (A.B.); caterina.marchio@ircc.it (C.M.); andrea.bertotti@ircc.it (A.B.); livio.trusolino@ircc.it (L.T.); alberto.bardelli@ircc.it (A.B.); claudio.isella@ircc.it (C.I.); 2Department of Oncology, University of Torino, 10060 Candiolo, Italy; alessia.corrado@unito.it; 3UT MD Anderson Cancer Center, University of Texas, Houston, TX 77030, USA; rminelli@mdanderson.org (R.M.); MDPeoples@mdanderson.org (M.D.P.); CABristow@mdanderson.org (C.A.B.); ACarugo@mdanderson.org (A.C.); 4Department of Medical Sciences, University of Torino, 10060 Candiolo, Italy

**Keywords:** CRC, synthetic lethality screening, pevonedistat, NEDD8, EGFR pathway

## Abstract

**Simple Summary:**

Identification of effective therapies for clinically aggressive, treatment-resistant colorectal cancer (CRC) remains an unmet clinical need. Targeted therapies against the epidermal growth factor receptor (EGFR) signaling axis lead to clinical benefits only in a small fraction of patients due to primary and acquired resistance. We previously showed that the NEDD8 pathway inhibitor pevonedistat induced tumor stabilization in preclinical models of aggressive CRC. Here, through synthetic lethality screenings, we found that pevonedistat could be successfully combined with EGFR pathway-targeted treatments in *BRAF*-mutant and *RAS-RAF* wild-type CRCs originally resistant to BRAF and EGFR blockade. We found that combined blockade of NEDD8 and EGFR pathways reverted compensatory feedback loops that reduced the efficacy of single treatments. Our results provide preclinical validation of a promising therapeutic strategy for clinically aggressive CRC resistant to EGFR and BRAF-targeted treatments.

**Abstract:**

Colorectal cancer (CRC) is a heterogeneous disease showing significant variability in clinical aggressiveness. Primary and acquired resistance limits the efficacy of available treatments, and identification of effective drug combinations is needed to further improve patients’ outcomes. We previously found that the NEDD8-activating enzyme inhibitor pevonedistat induced tumor stabilization in preclinical models of poorly differentiated, clinically aggressive CRC resistant to available therapies. To identify drugs that can be effectively combined with pevonedistat, we performed a “drop-out” loss-of-function synthetic lethality screening with an shRNA library covering 200 drug-target genes in four different CRC cell lines. Multiple screening hits were found to be involved in the EGFR signaling pathway, suggesting that, rather than inhibition of a specific gene, interference with the EGFR pathway at any level could be effectively leveraged for combination therapies based on pevonedistat. Exploiting both *BRAF*-mutant and *RAS/RAF* wild-type CRC models, we validated the therapeutic relevance of our findings by showing that combined blockade of NEDD8 and EGFR pathways led to increased growth arrest and apoptosis both in vitro and in vivo. Pathway modulation analysis showed that compensatory feedback loops induced by single treatments were blunted by the combinations. These results unveil possible therapeutic opportunities in specific CRC clinical settings.

## 1. Introduction

Molecular heterogeneity of colorectal cancer (CRC) is accountable for its significant variability in therapeutic regimen choice and treatment response [1,2]. For patients with metastatic CRC, targeted antibody therapy against epidermal growth factor receptor (EGFR) is recommended, since it was found to prolong the overall survival [3,4]. Nevertheless, genetic mechanisms of resistance include activating mutations at EGFR downstream signal transducers, such as *KRAS* and *BRAF*, that correlate with poor response to anti-EGFR antibodies. [5,6,7]. In patients with *BRAF*-mutant metastatic CRC, combined suppression of the EGFR–RAS–RAF–MAPK axis with cetuximab, encorafenib, and binimetinib displayed clinical efficacy [8]. Notably, even in the absence of known resistance mechanisms, EGFR-targeted therapy is not effective or durable in a non-negligible fraction of *RAS/RAF* wild-type (WT) cases [9,10]. Similarly, not all *BRAF*-mutant patients display good or long-lasting response to the combined treatment, which highlights an unmet clinical need for CRC.

We recently demonstrated that pevonedistat, by blocking conjugation of the ubiquitin-like protein NEDD8 to its targets [11,12,13], is particularly effective in preclinical models of poorly differentiated, clinically aggressive CRC [14]. Notably, response to pevonedistat is independent from the mutational status of the RAS/RAF axis and inversely associated with cetuximab sensitivity [14]. As a first-in-class agent, pevonedistat was employed in a series of phase I/II/III clinical trials alone [15,16] or in combination with chemo-/radiotherapy in both hematological and solid neoplasms [17,18]. However, in CRC patient-derived and cell line-derived xenografts, the best in vivo response observed was disease stabilization [14]. We therefore considered systematic exploration of combinations between pevonedistat and available therapeutic strategies as a valid approach to improve in vivo responses.

To identify drugs that might cooperate with NEDD8 inhibition, we performed an shRNA-based “drop-out” loss-of-function synthetic lethality screening on four CRC cell lines selected from a large collection [19] for known levels of sensitivity to pevonedistat and molecular features covering CRC heterogeneity [14]. We found and validated actionable candidates whose inhibition combined with pevonedistat significantly ameliorated the response in various CRC preclinical models, thus suggesting new therapeutic opportunities.

## 2. Results

### 2.1. Genetic Screening for Synthetic Lethality with Pevonedistat in CRC Cell Lines

Four CRC cell lines, all resistant to cetuximab, were selected from our collection based on our previously published data [14,19]. In particular, CAR1 cells are *RAS/RAF WT*, resistant to anti-EGFR treatments and sensitive to pevonedistat (Figure 1A). All three remaining lines have intermediate sensitivity to pevonedistat (40–80% cell viability at the 1 µM dose), with two of them (LIM2099 and SW403) harboring *KRAS* mutations and the third (WiDr) carrying *BRAF* mutation. We then defined optimal conditions for the synthetic lethality screenings (i.e., reduced growth rate under prolonged treatment without major cytotoxicity) by performing a long-term proliferation assay. Different pevonedistat concentrations significantly reduced—but did not abrogate—proliferation of the four CRC cells: 25 nM for CAR1, 100 nM for LIM2099 and SW403, and 200 nM for WiDr cells (Figure 1B).

For the synthetic lethality screening, we used a custom pooled, barcode-coupled shRNA library of 2000 shRNAs, covering about 200 genes that are targets of FDA-approved drugs (for library details, genes, corresponding pathway, and drugs available, see Appendix A). Genomic DNA was extracted from cell lines 96 h after library infection and then at the endpoint of the screening selection, which corresponded to the completion of 20 doublings in the absence and 10 doublings in the presence of pevonedistat, due to the slowing of cell growth caused by the drug treatment. Extracted DNA was processed by PCR amplification of integrated library constructs followed by NGS to perform quantification of each shRNA representation (Figure 1C; see also Appendix A). Synthetic lethal candidate genes were independently identified in each cell line for having at least four different shRNAs significantly depleted in all three screening replicates (Appendix A). Figure 1D shows the results for WiDr cells as an example. Overall, seventeen genes met our stringent selection, including confirmed basal expression in the screened cells (Appendix A). Notably, each cell line provided a different set of candidate genes, with no candidate shared between all cells (Table 1). With the remarkable exception of *BRAF* mutation in WiDr, candidate hit genes did not harbor any genetic alteration, e.g., point mutations or copy number alterations, and in 9 out of 17 cases, the shRNA dropout occurred in cells that displayed the highest level of expression of the target gene among selected cells (Appendix A).

### 2.2. Involvement of EGFR Pathway Genes in Pevonedistat Synthetic Lethality

The lack of a consistent overlap between hit genes among the four cell lines highlighted the absence of a highly recurrent pevonedistat synthetic lethal gene in CRC. However, the complex genomic makeup of cancer cells could strongly influence their sensitivity to the blockade of different components of the same pathway [9], which supports the strategy of targeting oncogenic pathways rather than a single gene to obtain a successful lethal combination in the heterogeneous setting of CRC. Accordingly, by conducting Ingenuity Pathway Analysis (IPA) [20], we identified three signaling cascades enriched in screening hits: ERK1/2, MAPK1, and AKT (Figure 2A).

STRING analysis [21] highlighted a strong protein functional interaction network (expressed as confidence) involving 12 hits and having EGFR and MAPK1 as a central hub (Figure 2B). Finally, Gene Ontology analysis on the 17 hit genes (Figure 2C) highlighted related biological processes controlling signal transduction (protein phosphorylation, MAPK cascade, regulation of protein stability, PI3-Kinase, intracellular signal transduction), proliferation (regulation of cell proliferation, cell division, and G2/M transition of mitotic cell cycle), and apoptosis (apoptotic process; for the complete list of biological functions, see Appendix A). Of note, *BRAF* was identified as a synthetic lethality hit in *BRAF*-mutant, BRAF blockade-resistant [22] WiDr cells, and *EGFR* was a hit in *RAS/RAF* WT, EGFR blockade-resistant CAR-1 cells.

Considering the recurrent involvement of the EGFR pathway through different hit genes in different cell lines, it is likely that synthetic lethality with pevonedistat arises by targeting this pathway in a tumor-specific way, depending on the molecular alterations leading to its oncogenic activation. Thus, we selected two main EGFR pathway screening hits, *BRAF* and *EGFR*, also in view of their therapeutic relevance for CRC.

### 2.3. Pevonedistat Cooperates with BRAF and EGFR Inhibition in BRAF-Mutant CRCs

Somatic *BRAF^V600E^* mutations occur in ~10% of CRCs; however, single-agent treatment with selective *BRAF^V600E^* inhibitors is ineffective in these patients [22]. This prompted successful clinical trials combining BRAF inhibitors with EGFR and MEK blockade. Nevertheless, primary resistance or early relapse are observed in many patients [8].

Results of our screening in *BRAF*-mutant WiDr cells suggested that NEDD8 inhibition should ameliorate response to *BRAF^V600E^* inhibition, and we reasoned that a vertical block on EGFR could further increase the observed cooperation between pevonedistat and the BRAF inhibitor, vemurafenib [22].

In short-term proliferation assays on WiDr cells, we observed a limited cooperation between pevonedistat and the *BRAF^V600E^* inhibitor vemurafenib (Appendix A). We reasoned that a short-term assay might not recapitulate the long-term culture conditions of the screening. We therefore performed 17 day colony formation assays and identified the minimum effective dose of pevonedistat on WiDr cells as 100 nM (Appendix A). Concordantly with its *BRAF*-mutant background, we confirmed that WiDr cells still retained colony formation potential in response to vemurafenib. Pevonedistat slightly but significantly increased the anti-proliferative effects of vemurafenib (*p* ≤ 0.05; Appendix A), compatible with the ascertained shRNA depletion in the screening. Moreover, we detected reduced clonogenic potential under the combined treatment versus vemurafenib alone (*p* ≤ 0.05; Appendix A), in line with the previously observed pro-apoptotic activity of pevonedistat on CRC cell lines [14]. We then confirmed in colony-forming assays that WiDr cells were resistant to cetuximab (Figure 3A–C) but displayed a slightly increased response to the vemurafenib plus cetuximab combination vs. vemurafenib alone (Figure 3B,C).

Interestingly, the triple combination of pevonedistat, vemurafenib, and cetuximab significantly reduced both the relative growth (Figure 3B) and the clonogenic potential (Figure 3C), thus ameliorating the response in terms of anti-proliferative and pro-apoptotic effects (for a complete statistical comparison among treatments, see Appendix A).

The efficacy of this combinatorial regimen on *BRAF*-mutant CRC was confirmed in an additional *BRAF*-mutant CRC cell line (SNUC5), in which inclusion of pevonedistat significantly reduced relative growth (*p* ≤ 0.01) and number of colonies (*p* ≤ 0.01) induced by vemurafenib plus cetuximab (Appendix A).

The promising data obtained in vitro set the basis for testing efficacy of the new therapeutic combination in vivo. WiDr cells were transplanted subcutaneously in NOD-SCID mice and treated with the standard vemurafenib plus cetuximab combination [22], with pevonedistat alone, or with the triplet. As expected, vemurafenib plus cetuximab as well as pevonedistat alone reduced tumor growth by about 15% and 50%, respectively, without inducing tumor regression (Figure 3D). In vivo, the triplet significantly blocked tumor growth of WiDr cells compared to vehicle-treated mice (*p* ≤ 0.001 at their sacrifice), vemurafenib plus cetuximab (*p* ≤ 0.001), or pevonedistat alone (*p* ≤ 0.01) at six weeks. Most importantly, the triple combination induced detectable tumor regression (tumor volume reduction of ~20%).

We finally took advantage of an additional in vitro model closer to the actual human tumor: a *BRAF*-mutant patient-derived organoid (PDO) named CRC0079. We evaluated the efficacy of the triplet to suppress organoids viability (Figure 3E and Appendix A), and, in parallel, we quantified caspase-3/7 activity both at 96 h and 24 h of treatment (Figure 3F and Appendix A). Pevonedistat significantly increased the effects of the vemurafenib plus cetuximab combination on proliferation at 96 h in a dose-dependent manner (Figure 3E). Moreover, a significant increase of apoptosis induced by pevonedistat was detected when combined with vemurafenib plus cetuximab treatment already at 24 h (Appendix A) and further increased at 96 h (Figure 3F), thus supporting improved anti-proliferative and pro-apoptotic effects of the triplet.

These results show that concomitant blockade of EGFR and NEDD8 pathways has the potential to improve treatment of patients with *BRAF*-mutant CRC.

### 2.4. Combined Effect of NEDD8 and EGFR Pathway Blockade in RAS/RAF WT CRCs

As previously mentioned, *EGFR* was identified as a hit gene in CAR1 cells that are *RAS/RAF* WT but almost totally resistant to cetuximab [19] by an unknown mechanism, a condition that is not infrequent in CRC patients [9]. Considering that dual blockade of EGFR and ERBB2 was previously found to increase efficacy in CRC patients [23], we tested the combined action on CAR1 cells of pevonedistat with either cetuximab or the dual EGFR/ERBB2 kinase inhibitor lapatinib. Using an area under the dose-response curve (AUC) analysis of short-term (96 h) proliferation assays, we found that lapatinib was more effective than cetuximab in reducing cell viability when combined with pevonedistat (AUC, pevonedistat alone = 0.722; plus cetuximab = 0.529, *p* = 6.78 × 10^−5^; plus lapatinib = 0.455, *p* = 2.13 × 10^−5^). For the long-term colony assay, CAR1 cells were therefore treated with pevonedistat or lapatinib alone and with their combination (Figure 4A). While pevonedistat alone significantly reduced both relative growth and number of colonies (Figure 4B,C), lapatinib alone only induced minor effects on colony formation (Figure 4C). Notably, their combination significantly further reduced both relative growth and number of colonies with respect to control and to each single treatment (Figure 4B,C).

Similar results were obtained in an additional CRC cell line, HCA7, *RAS/RAF* WT, but this was only partially sensitive to cetuximab [19]. Additionally, in this case, lapatinib displayed better performances than cetuximab when combined with pevonedistat in colony assay (Appendix A). The lapatinib plus pevonedistat combination significantly reduced both relative growth and number of colonies (Figure 4D,E,F), consistent with a crucial role of EGFR blockade in the lethal response to NEDD8 inhibition.

We then evaluated the efficacy of pevonedistat and EGFR inhibition in vivo on HCA7 xenografts, since our preliminary experiments indicated variable engraftment and growth of CAR1 cells. Indeed, while lapatinib did not exert detectable effects, pevonedistat markedly reduced tumor growth (Figure 4G), however, without inducing tumor stabilization (Figure 4H). Strikingly, the doublet completely abrogated tumor growth (Figure 4G), thus inducing consistent stabilization (Figure 4H) and supporting a therapeutic potential of this combinatorial regimen in *RAS/RAF* WT, cetuximab-resistant CRC.

### 2.5. Multiple Mechanisms of Cooperation between Pevonedistat and EGFR Inhibitors

We sought to address the molecular mechanism underlying the combined action of pevonedistat and EGFR inhibition in *BRAF*-mutant and *RAS/RAF* WT CRC.

Western blot analysis of *BRAF*-mutant WiDr cells (Figure 5A) showed that pevonedistat induced strong upregulation of Tyr1068-phosphorylated EGFR (p-EGFR), associated with a more modest upregulation of the downstream Ser473-phosphorylated AKT (p-AKT) and the T202/Y204-phosphorylated ERK (p-ERK), indicating a compensatory mechanism leading to EGFR pathway upregulation in response to pevonedistat. Notably, it was reported that activated EGFR stimulated receptor neddylation, which enhanced its subsequent ubiquitylation and degradation [24]; thus, p-EGFR could be stabilized by pevonedistat treatment through blockade of its NEDD8-mediated ubiquitylation. Interestingly, vemurafenib and cetuximab completely abrogated pevonedistat-induced p-EGFR and p-AKT upregulation, although this poorly affected their basal level. Overall, these data indicate that the action of pevonedistat on WiDr cells was mitigated by a compensatory positive rebound on the EGFR pathway, which was abrogated by concomitant EGFR and BRAF blockade. These observations were confirmed by immunohistochemistry analysis of tumor sections derived from WiDr xenografts at the end of treatments. In fact, the triplet vemurafenib + cetuximab + pevonedistat induced a striking depletion of Ki67, which supports a strong reduction of the proliferative rate without additional effects in combinatorial arms for p21 and p27 protein expression (Figure 5B and Appendix A).

On top of this, pevonedistat was found to induce a strong increase of γH2Ax and p21 levels (Figure 5A and Appendix A), indicating activation of the DNA damage response and proliferative block, known effects of NEDD8 pathway inhibition [14,25]. Such upregulation was not induced by vemurafenib or cetuximab or their combination, nor was it modified by these drugs when combined with pevonedistat (Figure 5A and Appendix A).

Western blot analysis was also performed in *RAS/RAF* WT CAR1 and HCA7 cells (Figure 5C,D and Appendix A). In CAR1 cells, at odds with the *BRAF*-mutant cell line, pevonedistat did not affect phosphorylation of EGFR or its downstream transducers but significantly increased p21 and γH2Ax protein levels (Figure 5C). Instead, lapatinib inhibited EGFR activation by reducing p-EGFR, p-AKT, and p-ERK (Figure 5C and Appendix A). In HCA7 cells, pevonedistat only slightly increased the levels of p-EGFR but significantly reduced p-AKT levels that instead were not affected by lapatinib alone or in combination (Figure 5D and Appendix A). p-ERK levels were not modified by pevonedistat or lapatinib (Figure 5D). Concordantly with our previous observations, p21 and γH2Ax were highly increased by pevonedistat (Figure 5C,D and Appendix A) and were not affected by lapatinib alone or in combination. IHC on tumor sections of HCA7 xenografts treated with pevonedistat, lapatinib, or the combination did not reveal significant additive effects on the abundance of Ki67-positive cells (Figure 5E and Appendix A), and there was a large increase in p21 and p27 positivity, mostly adjacent to necrotic areas. These results suggest some levels of selection during treatment and are in line with the observed in vivo tumor stabilization (Figure 4H).

Overall, these data show that the improved efficacy of concomitant inhibition of the NEDD8 and the EGFR pathways results, in part, from parallel actions with DNA damage response induced by pevonedistat and proliferative arrest induced by EGFR axis inhibition and, in part, from abrogation of reciprocal compensatory signaling feedbacks on the EGFR pathway.

## 3. Discussion

Our screening for synthetic lethality with pevonedistat in CRC cell lines highlighted multiple candidate drug target genes that displayed limited nominal overlap across cell lines but consistent convergence on the EGFR pathway, irrespective of the *KRAS/BRAF* mutational status. Indeed, CRC is typically dependent on this pathway, even in the absence of its aberrant activation by genetic alterations [9,26]. Our findings indicated that combined administration of pevonedistat and targeted therapies against the EGFR pathway led to increased anti-proliferative and pro-apoptotic activity both in vitro and in vivo.

Indeed, independently of the CRC model used, pevonedistat invariantly promoted upregulation of p21 and γH2Ax protein levels, thus inducing DNA damage and cell cycle arrest [14] and confirming the general mechanism of action impinging on re-replication and the CRL4–CDT2–SET8/p21 axis [27,28]. However, depending on the CRC genetic makeup, we found that the synergy between pevonedistat and EGFR pathway blockade took place through partially distinct targets and regulatory mechanisms.

In *BRAF^V600E^* WiDr cells, pevonedistat induced adaptive phosphorylation of EGFR and of its downstream transducer AKT, thus leading to a compensatory survival and proliferation signal. A similar effect was already described in other tumor models, highlighting a general mechanism by which pevonedistat efficacy can be limited [29,30]. Notably, EGFR activation was also previously described upon treatment with BRAF inhibitors [22], leading to clinical application of combined BRAF/EGFR blockade in *BRAF*-mutant CRC [8]. This may explain the observed in vivo synergism in *BRAF*-mutant CRC when NEDD8 inhibition is associated with complete EGFR pathway blockade by vemurafenib plus cetuximab.

Differently, in *RAS/RAF* WT models resistant to EGFR inhibition, pevonedistat had no major effects on the EGFR pathway, as previously reported for glioblastoma [31] and human myeloma [32]. In this case, the cooperation between pevonedistat and EGFR pathway blockade could be ascribed to parallel actions leading to cell death and inhibition of proliferation.

Overall, treatment of *BRAF*-mutant and *RAS/RAF* WT CRC respectively with a triple (pevonedistat, vemurafenib, and cetuximab) or a double (pevonedistat and lapatinib) combination, leading to effective concomitant targeting of the NEDD8 and the EGFR pathways, led to substantial improvements with respect to single pathway blockade, where no current therapeutic options seem effective.

Although the CRC molecular heterogeneity led to partially distinct mechanisms of cooperation between NEDD8 and EGFR pathway blockade, the observed preclinical results warrant further investigations to better identify patients that could benefit from the various possible versions of the combined blockade.

## 4. Materials and Methods

### 4.1. Cell Lines and Drugs

CAR1 (JCRB—Japanese Collection of Research Bioresources Cell Bank), HCA7 (ECACC—European Collection of Authenticated Cell Cultures), LIM2099 (kindly provided by Dr. Walkerl), SW403 (ATCC—American Type Culture Collection), WiDr (kindly provided by Prof. Bernards), and SNUC5 (KCLB—Korean Cell Line Bank) CRC cell lines were maintained in their respective media as recommended by suppliers. Cells were screened for mycoplasma contamination using the Venor GeM Classic kit (Minerva Biolabs). The identity of each cell line was checked by Cell ID System and by Gene Print 10 System (Promega, Madison, WI, USA).

Pevonedistat (MLN4924) was purchased from Active Biochem. Cetuximab was obtained from Merck and lapatinib from Sequoia Research Products. Vemurafenib (PLX4032 for in vitro and PLX4720 for in vivo experiments) was acquired from Selleckchem Chemicals.

### 4.2. shRNA Library Design and Construction

A custom library of 2000 shRNAs focused on genes target of FDA-approved drugs was constructed by using chip-based oligonucleotide synthesis and was cloned into a pRSI-U6-(sh)-UbiC-GFP-2A-Puro lentiviral vector containing the puromycin-resistance and the GFP fluorescent marker (Cellecta Inc., Mountain View, CA, USA) according to published protocol [33]. For library details, cells transduction and screening, and sequence and data analysis, see Appendix A.

### 4.3. Enrichment Analysis of Candidate Hits

Ingenuity Pathway Analysis (IPA) for network identification and Gene Ontology (GO) analysis were performed on 17 candidate hits. Functional annotations of biological processes (BP) were considered when they scored with a *p*-value ≤ 0.05 and at least 3 terms were included. Moreover, protein interaction network was analyzed via online Search Tool for the Retrieval of Interacting Genes (STRING: https://string-db.org (accessed on 1 May 2020)), and confidence among genes was considered with a minimum required interaction score of 0.400.

### 4.4. In Vitro Drug Efficacy Studies

For short-term viability assays, CRC cell lines were seeded at 1 × 10^3^ cell/well density in complete growth medium in 96-well plastic culture plates at day 0. The following day, serial dilutions of drugs were added to the cells in additional serum-free medium at concentrations between 10 nM and 5000 nM in diverse cell lines. After 96 h treatment, cell viability was assessed by measuring ATP content through Cell Titer-Glo Luminescent Cell Viability assay (Promega). Luminescence was measured by Perkin Elmer Victor X4. Specifically, after 96 h of pevonedistat treatment, cells were defined as sensitive when average cell viability between 500 nM and 1 µM was lower than 40%, while they were considered resistant with cell viability higher than 80% at 1 µM and higher than 50% at 5 μM. Cell lines that were not strictly included in these rules were indicated to have an intermediate sensitivity to pevonedistat. For long-term proliferation assays, 5 × 10^4^ CRC cells were seeded in complete growth medium in 6-well plates. Pevonedistat was added as single administration at increasing concentrations in triplicates for all the cell lines tested (25, 50, 100, 200, and 400 nM). For colony formation assays, WiDr cells were treated with pevonedistat (100 nM), cetuximab (20 μg/mL), and vemurafenib (125 nM). CAR1 cells were treated with pevonedistat (25 nM) and lapatinib (250 nM). HCA7 cells were treated with pevonedistat (50 nM) and lapatinib (30 nM). After 10 to 17 days of culture, depending on cell line, colonies were fixed, stained with crystal violet, and quantified by using ImageJ analysis software for the evaluation of the number and the colonies growth (expressed as relative growth).

### 4.5. Organoids Drug Efficacy Study

*BRAF*-mutant patient-derived organoid (PDO) CRC0079 was cultured inside a Cultrex growth factor reduced BME type 2 (R & D systems) drop in DMEM/F-12 HAM supplemented with B-27 and N-2 (both from GIBCO), 1 mM of N-acetylcysteine, penicillin-streptomycin, 2 mM of L-glutamine, and EGF at a final concentration of 20 ng/mL (all from SIGMA-ALDRICH) and was grown in a 37 °C and 5% CO_2_ air incubator as previously described [34]. For in vitro assays, PDO CRC0079 was mechanically disaggregated, trypsinized, and plated at 2.5 × 10^3^ cell/well density in standard growth medium (without EGF and supplemented with 2% BME) in 96-well culture dishes coated with BME at day 0. After 48 h, serial dilutions of pevonedistat (150, 300, and 500 nM), vemurafenib (125 nM), and cetuximab (20 µg/mL) were added. After 24 h and 96 h of treatment, cell viability was determined by measuring ATP content through CellTiter-Glo^®^ Luminescent Cell Viability assay (Promega), and caspase-3/7 activity was detected through Caspase-Glo^®^ 3/7 Assay System (Promega). Luminescence was measured by Perkin Elmer Victor X4. Caspase-3/7 relative activity was normalized versus CellTiter-Glo values. Images were acquired at Cytation 3 Cell Imaging Multi-Mode Reader (BioTek Instruments, Inc., Winooski, VT, USA).

### 4.6. Animals and Ethics Statement

Non-obese diabetic/severe combined immunodeficiency (NOD/SCID) mice were purchased from Charles River. Only female mice 6–12 weeks old (15–20 g weight) were used for experimental procedures. Investigation was conducted in accordance with the ethical standards and according to national and international guidelines. In vivo studies were performed after approval from our fully authorized animal facility, notification of the experiments to the Ministry of Health (as required by the Italian Law) (IACUCs No. 195/2015-PR), and in accordance with EU directive 2010/63.

### 4.7. In Vivo Studies

HCA7 and WiDr CRC cell lines were subcutaneously transplanted in the flank of the immunocompromised NOD-SCID mice (5 × 10^6^ cells per mouse) and resuspended in Corning^®^ Matrigel^®^ Growth Factor Reduced (GFR) Basement Membrane Matrix (Life Science) 1:1 in PBS 1X. When the tumor volume reached approximately 250–300 mm^3^, animals were randomized in the experimental groups of treatments. HCA7 xenografts were treated either subcutaneously with pevonedistat 90 mg/kg (dissolved in 20% of 2-hydroxypropyl-β-cyclodextrin/saline) BID/BIW, or by oral gavage with lapatinib 100 mg/kg (dissolved in saline) daily or with the combination of the two drugs for six weeks. WiDr xenografts were assigned to treatment with different regimens for six weeks: (1) pevonedistat (90 mg/kg) BID/BIW subcutaneously; (2) cetuximab (20 mg/kg) intraperitoneally twice a week plus vemurafenib (PLX4720: 60 mg/kg dissolved in 0.2% Tween 80 and 1% methylcellulose in sterile H_2_O) daily by oral gavage; (3) a triple drugs combination (cetuximab plus vemurafenib plus pevonedistat) in which each compound was administered at the same dose and scheduled as single agents. Tumor size was evaluated weekly by caliper measurements, and the approximate volume was calculated using the formula 4/3π · (d/2)2 · D/2, where d is the minor tumor axis and D is the major tumor axis.

### 4.8. Immunoblot Analysis

The biochemical responses of cells treated with drugs were analyzed by Western blot. Cells were plated in complete culture medium and treated with pevonedistat (0.5 µM for CAR1 and 1 µM for HCA7 and WiDr), lapatinib (0.25 µM for CAR1 and 1 µM for HCA7), cetuximab (20 μg/mL for WiDr), vemurafenib (1 µM for WiDr), or their combinations for 24 h (for WiDr and CAR1) or 48 h (for HCA7). The lysates were then collected using Lysis Buffer (Tris-HCL 1M Ph 6.8 and SDS 10% in H_2_O) at 95 °C. All lysates were prepared and resolved by SDS gel electrophoresis and probed with the following primary antibodies: p-EGFR (Y1068) (Abcam), EGFR (Enzo Life Science), p-ERK1/2 (T202/Y204), ERK1/2, p-AKT (S473), AKT1/2, p21, and γH2Ax (all from Cell Signaling Technology). Vinculin was used as a normalizer. Membranes were then incubated with the appropriate secondary antibody linked to horseradish peroxidase. Images were cropped at specific protein band of interest to improve the clarity of data presentation. Protein quantification from Western blot was performed by using ImageJ software. The level of the phosphorylated forms was expressed as ratio to the respective total form after normalization of each sample loading by using vinculin.

### 4.9. Immunohistochemistry

Tumor fragments obtained upon transplantation of WiDr and HCA7 CRC cells after different therapeutic regimens were formalin-fixed and paraffin-embedded. For experimental details and antibodies, see Appendix A.

### 4.10. Statistical Analysis

Data are represented as mean ± standard deviation (SD) of biological triplicates. Comparisons of drug efficacy among groups were performed using one-way ANOVA followed by Student’s *t*-test for unpaired samples. *p* ≤ 0.05 was considered significant.

## 5. Conclusions

Our study provides a preclinical validation of a promising therapeutic strategy for clinically aggressive CRC resistant to EGFR and BRAF-targeted treatments. We suggest that the combination of pevonedistat and EGFR inhibitors in *BRAF*-mutant and *RAS-RAF* wild-type CRCs is a successful approach to abrogate compensatory feedback loops and increase the efficacy of standard treatments.

## Figures and Tables

**Figure 1 cancers-13-03805-f001:**
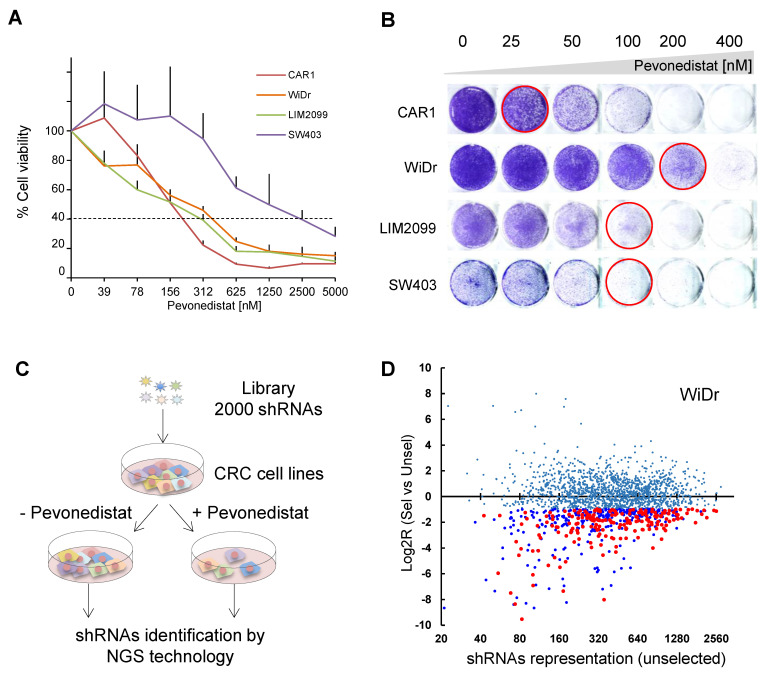
Genetic screening for synthetic lethality with pevonedistat in CRC cell lines. (**A**) Four CRC cell lines were treated with increasing concentrations of pevonedistat for 96 h, and short-term cell viability was determined by using Cell Titer Glo. Response of each cell line is reported as percentage of cell viability for *n* = 3 experiments (mean ± SD). (**B**) Long-term proliferation assay was performed for *RAS/RAF WT* CAR1, *BRAF*-mutant WiDr, and KRAS-mutant LIM2099 and SW403 cells in the presence of increasing concentrations of pevonedistat for 8 to 15 days. Plates were fixed with crystal violet. (**C**) Schematic representation of the in vitro drop-out shRNAs screening performed in the aforementioned CRC cell lines is reported. Each cell line was infected in three independent experiments with the library composed of 2000 shRNAs targeting druggable genes and then treated (+) or not (−) with pevonedistat for about 20 doublings in the absence and 10 doublings in the presence of the drug. Cell were harvested, and the library was amplified by PCR and sequenced by NGS technology. (**D**) Representative scatter plot of one screening in WiDr cells. The x-axis reports the abundance of each shRNA in unselected cells, expressed as counts per million; the y-axis reports the LOG_2_ ratio (R) of shRNA representation in selected vs. unselected cells in the same screening. Light blue dots: not significant depleted shRNAs; dark blue dots: significant depleted shRNAs in one replica experiment; red dots: significant depleted shRNAs in triplicate experiments.

**Figure 2 cancers-13-03805-f002:**
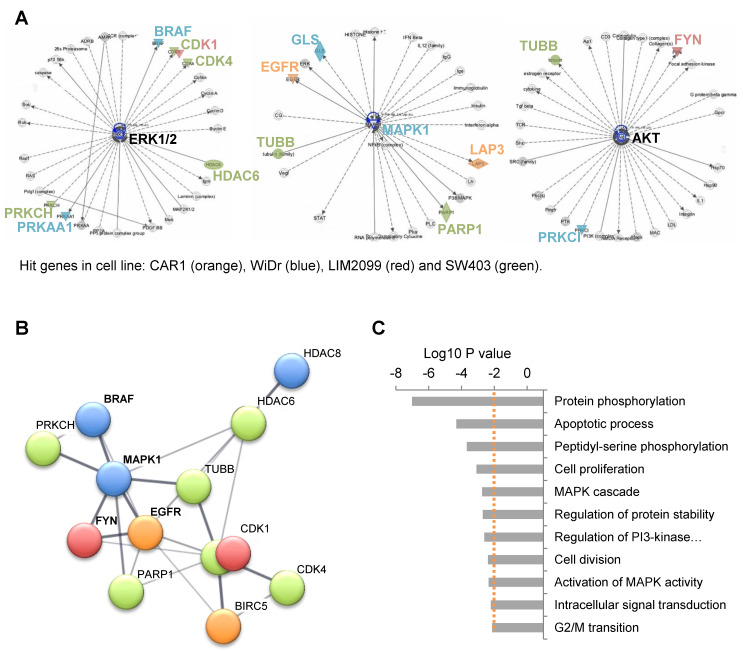
Involvement of oncogenic EGFR pathway due to pevonedistat treatment in CR(C). (**A**) IPA was performed on the 17 hit genes to analyze their involvement in functional networks. In the three networks emerged from the analysis, candidate genes are highlighted by colors representing the screening cell line of origin. (**B**) Gene interaction network generated by STRING, including 12 of the 17 candidates. Line thickness represents strength of data confidence (interaction score > 0.400). Color nodes as above. (**C**) Result of Gene Ontology biological processes (GO-BP) analysis performed with DAVID on the hit genes. Representative biological functions are reported and ranked by LOG_10_
*p*-value (for a complete list of significant functions, see Appendix A).

**Figure 3 cancers-13-03805-f003:**
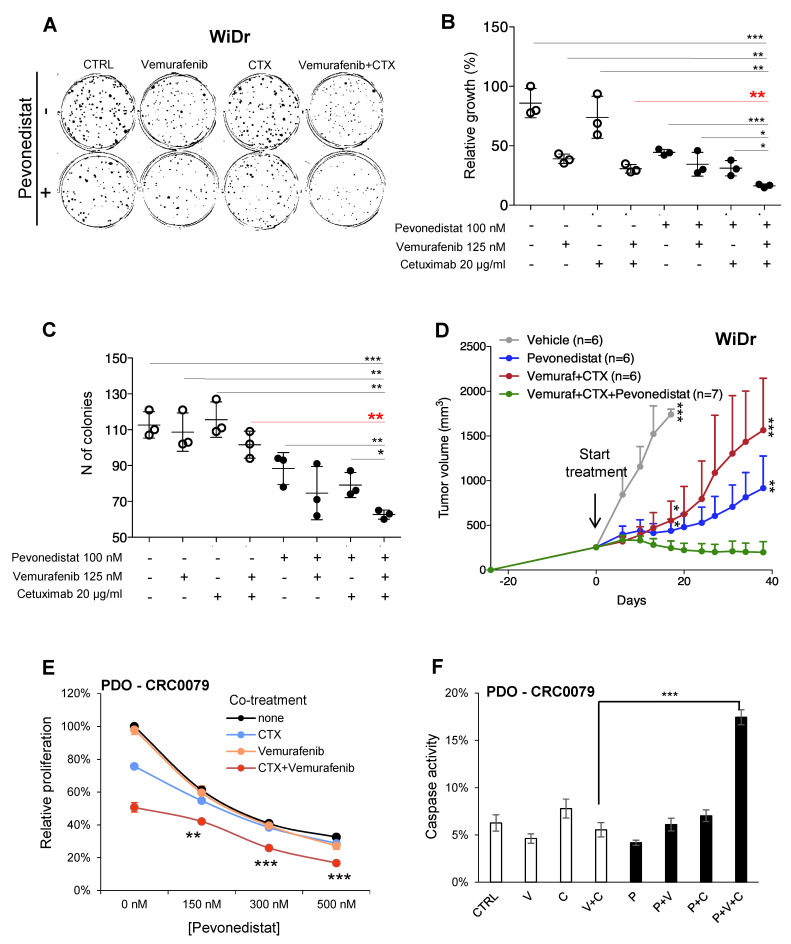
Pevonedistat and EGFR pathway cooperate in *BRAF*-mutant CRCs. (**A**) Clonogenic potential of WiDr cells was evaluated in response to pevonedistat (100 nM), cetuximab (CTX, 20 µg/mL), and vemurafenib (125 nM) alone or in combination. After 17 days, cells were fixed by crystal violet. (**B**,**C**) Relative growth of colonies (**B**) and their number (**C**) were quantified in triplicate. Significant differences among triple combination and other treatment groups (pevonedistat—P; vemurafenib—V; cetuximab—C) were analyzed by applying Student *t* test (*: *p* < 0.05; **: *p* < 0.01; ***: *p* < 0.001; the red color highlights comparisons between the triple treatment and the standard double treatment). (**D**) Mice transplanted with WiDr cells were randomized in four groups and treated with vehicle, pevonedistat (90 mg/kg), cetuximab (CTX—20 mg/kg), and vemurafenib (60 mg/kg) alone or in combination. Tumor volume (mean ± SD) was measured weakly. Growth curves were compared by Student *t* test analysis for the evaluation of significant differences among groups (***: *p* < 0.001; **: *p* < 0.01; *: *p* < 0.05). (**E**,**F**) *BRAF*-mutant patient-derived organoid (PDO, CRC0079) was treated for 96 h with vemurafenib (125 nM), cetuximab (CTX, 20 µg/mL) and increasing pevonedistat concentrations (150, 300, 500 nM). Data are expressed as average ± SD of three technical replicates. (**E**) Cells proliferation was quantified by Cell Titer Glo assay, and significant differences among groups were calculated by applying Student *t* test (V + C vs V + C + P: **: *p* < 0.01; ***: *p* < 0.001). (**F**) Caspase-3/7 activity was quantified by Caspase Glo Assay after 96 h treatment and expressed as ratio of proliferation. Significant differences among V + C and V + C + P groups were calculated by applying Student *t* test (***: *p* < 0.001).

**Figure 4 cancers-13-03805-f004:**
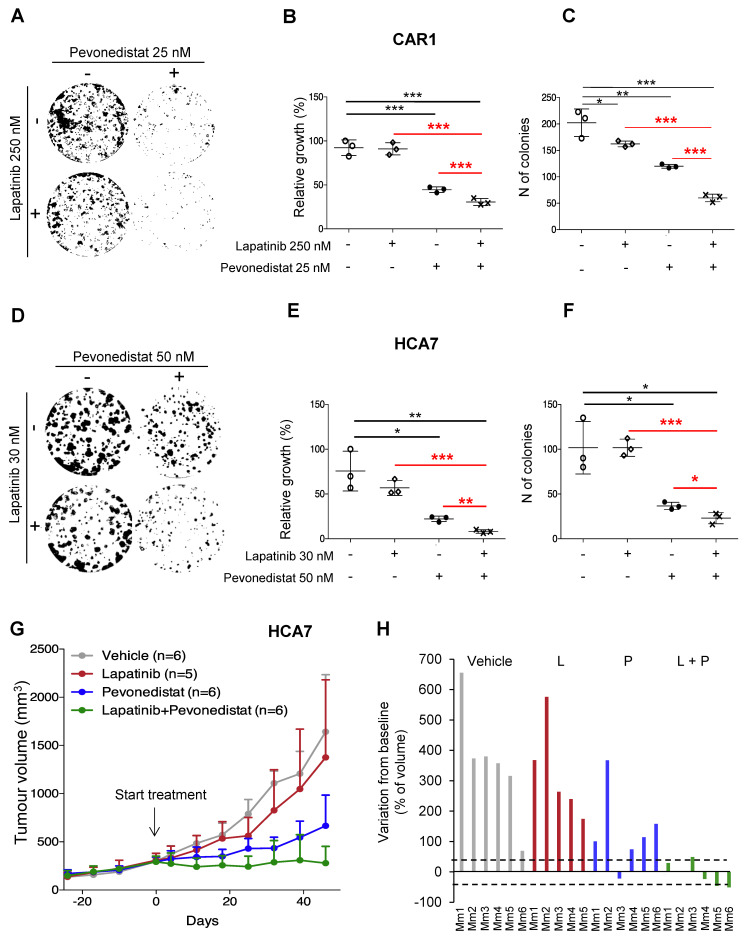
Combination of pevonedistat and EGFR pathway inhibition in *RAS/RAF* WT CRCs. (**A**) Colony formation of CAR1 cells was evaluated in response to pevonedistat (25 nM), lapatinib (250 nM), and combinatorial treatment of both. After 10 days cells were fixed by crystal violet. (**B**,**C**) Relative growth of CAR1 colonies (**B**) and their number (**C**) were quantified in triplicate as shown by dot plot. Significant differences among triple combination and other treatment groups (pevonedistat—P; lapatinib—L) were analyzed by applying a Student *t* test (*: *p* < 0.05; **: *p* < 0.01; ***: *p* < 0.001; the red color highlights comparisons between combined and single treatments). (**D**) Colony formation of HCA7 cells was evaluated in response to pevonedistat (50 nM), lapatinib (30 nM), and combinatorial treatment of both. After 13 days, cells were fixed by crystal violet. (**E**,**F**) Relative growth of HCA7 colonies (**E**) and their number (**F**) were quantified in triplicate, as shown by the dot plot. Significant differences among triple combination and other treatment groups (pevonedistat—P; lapatinib—L) were analyzed by applying a Student *t* test (*: *p* < 0.05; **: *p* < 0.01; ***: *p* < 0.001; the red color highlights comparisons between combined and single treatments). (**G**) Mice transplanted with HCA7 cells were treated for around 7 weeks with single or combinatorial agents (pevonedistat 90 mg/kg; lapatinib 100 mg/kg). Tumors were measured weekly, and growth was reported as mean ± SD. (**H**) Waterfall plot for treatment response compared with tumor volume at baseline in each mouse (Mm) of respective group of treatment. The dotted line represents the cut-off values for therapy response indicated as >35% for progression and 35 ≤ % ≤ −50 as tumor stabilization.

**Figure 5 cancers-13-03805-f005:**
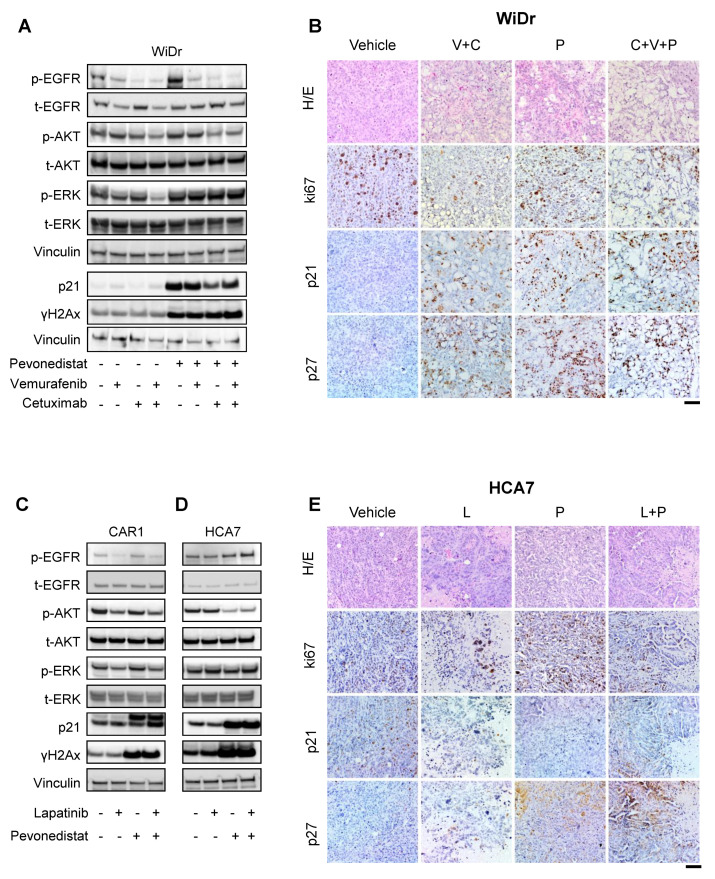
Characterization of *BRAF*-mutant and *RAS/RAF* WT CRC response to pevonedistat and EGFR inhibition. (**A**) Western blot was performed on lysates from WiDr cells treated with pevonedistat (1 µM), cetuximab (20 µg/mL), vemurafenib (1 µM), or their combinations for 24 h. EGFR, AKT, and ERK in their phosphorylated (p-) or total (t-) forms were evaluated. P21 and γH2Ax were analyzed in the aforementioned conditions. Vinculin was used as a normalizer. The uncropped Western blots have been shown in Appendix A. (**B**) Tumors from WiDr cells were treated with vehicle, pevonedistat (90 mg/kg), cetuximab (20 mg/kg), and vemurafenib (60 mg/kg) alone or in combination. Tumor sections after drug treatments (V: vemurafenib; C: cetuximab; P: pevonedistat) were analyzed by EE and IHC for proliferative marker (ki67, p21, and p27). Representative images were acquired at 10× magnification. Scale bar: 200 µm. (**C**,**D**) Western blot was performed on lysates obtained from drug-treated *RAS/RAF* WT CRC cells CAR1 (**C**) and HCA7 (**D**). CAR1 were treated with pevonedistat (0.5 µM), lapatinib (0.25 µM), or their combinations at 24 h after drugs treatment (**C**). HCA7 were analyzed upon 48 h treatment with pevonedistat (1 µM), lapatinib (1 µM), or their combinations (**D**). EGFR, AKT, and ERK in their phosphorylated (p-) or total (t-) forms were evaluated. p21 and γH2Ax protein levels were evaluated in the aforementioned conditions. Vinculin was used as a normalizer. (**E**) HCA7 transplanted mice were treated with pevonedistat (90 mg/kg) and lapatinib (100 mg/kg) or their combination. Tumor sections after drug treatments (L—lapatinib; P—pevonedistat) were processed and analyzed as described in panel (**B**).

**Table 1 cancers-13-03805-t001:** Candidate hit genes identified by screening for synthetic lethality with pevonedistat. Four CRC cell lines with diverse mutational status were independently screened for library selective shRNA drop-out upon treatment with pevonedistat. Candidate genes (+) are reported for each cell line in alphabetical order.

Mutational Status	CAR1	WiDr	LIM2099	SW403
	WT	BRAFm	KRASm	KRASm
Candidate Genes	3	6	2	7
BIRC5	+	−	−	−
EGFR	+	−	−	−
LAP3	+	−	−	−
BRAF	−	+	−	−
GLS	−	+	−	−
HDAC8	−	+	−	−
MAPK1	−	+	−	−
PRKAA1	−	+	−	−
PRKCI	−	+	−	−
CDK1	−	−	+	+
FYN	−	−	+	−
CDK4	−	−	−	+
HDAC6	−	−	−	+
PARP1	−	−	−	+
PGD	−	−	−	+
PRKCH	−	−	−	+
TUBB	−	−	−	+

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
