# Peer review of "Synthetic Lethality Screening Highlights Colorectal Cancer Vulnerability to Concomitant Blockade of NEDD8 and EGFR Pathways"

_cancers, 2021, doi:10.3390/cancers13153805_

Round 1

Reviewer 1 Report

In this work the Authors investigated the molecular basis of pevonedistat mechanism of action in CRC cell lines, highlighting that “…multiple candidate drug target genes... displayed limited nominal overlap across cell lines but consistent convergence on the EGFR pathway, irrespective of the KRAS/BRAF mutational status”. They also demonstrated that combined administration of pevonedistat and targeted therapies against the EGFR pathway leads to increased anti-proliferative and pro-apoptotic activity. Finally, they have suggested that concomitant NEDD8 and EGFR pathways inhibition, has led to improvements in anticancer activity respect to single pathway blockade.

In my opinion the topic of great scientific interesting, the experimental design is well conducted, the results of the experiments performed support the conclusions and the paper has been clearly written.

Author Response

We thank the Reviewer for the positive evaluation.

Reviewer 2 Report

The study.

The authors previously found that the NEDD8-activating enzyme inhibitor pevonedistat induced tumor stabilization in preclinical models of poorly differentiated CRC. Though, only a reduced number of cell lines (13%) was sensible to this drug. In the present study, they aimed to identify drugs that could be combined with pevonedistat, to transform tumor stabilization into a real regression. They used a “drop-out” loss-of-function synthetic lethality screening by a shRNA library covering 200 drug-target genes in four different CRC cell lines (CAR1, LIM2099, SW403, WiDr). This very interesting approach, did not produce substantial results, as the synthetic lethal gene testing showed no overlap between hit genes among the four cell lines tested. Of course, the chance of obtaining significant results on such a limited sampling, compared with the previous work of the same group on 122 CRC cell lines (https://doi.org/10.1093/jnci/djw209), was very limited. Thus, the authors used a generic computational analysis to justify a switch of the study on the EGFR pathway. This switch also produced the need of introducing other cell lines to validate results (SNUC5, not reported in materials and methods, HCA7) and of a BRAF-mutated organoid culture (CRC0079). Despite this break between the first and second part of the study, results are encouraging and suggest the feasibility of pevonedistat + EGFR pathway inhibitors combination in selected CRC patients. Though, some results are oversimplified and should be improved by additional tests and conceptual corrections.

Comments and suggestions.

  • Lines 49-50: EGFR therapy shows some efficacy only against metastatic CRC, this should be clearly stated in the introduction

  • Line 93: why controls and pevonedistat-treated cells have been collected after such different doubling times (10 vs 20 doublings)? This control could be intrinsically leaky and doesn’t allow a trustable comparison. Please provide data in a homogeneous setting

  • In CAR1 cells, EGFR is apparently not necessary and in their previous paper the authors showed that CRC cells sensitive to NEDD8 are those not relying on EGFR activation. The authors should knock-out EGFR and/or Her2 from CAR1 cells and show if the lapatinib+ pevonedistat combination is still working in synergy. As a simplified alternative, the authors could repeat the growth, colony formation and WB tests in the presence/absence of exogenous EGF.

  • In WB methods (lines 459-460) is stated that signals were evaluated after 24h (for WiDr and CAR1) or 48h (for HCA7). This is not justified and I invite the authors to repeat the test for WiDr and CAR1 at 48h, and/or for HCA7 at 24h, and compare the results at the same time points. As Erk1-2 is a doublet of p42 and p44 proteins and only MAPK1 (=Erk2) has been involved in Pevonedistat activity, please run the blots in 8% gels and enough so that the doublet is well separated and the specific modulation of Erk2 can be judged.

  • Lines 274-278: Western blot analysis of BRAF-mutant WiDr cells (Figure 5A) showed that pevonedistat induces strong up-regulation of Tyr1068-phosphorylated EGFR (…omissis…) indicating a compensatory mechanism leading to EGFR pathway upregulation in response to pevonedistat. Activated EGFR is a known target of NEDD8-mediated ubiquitylation and down-regulation (https://doi.org/10.1074/jbc.m513034200), thus p-EGFR could be simply stabilized by Pevonedistat treatment. This is a known mechanism, and should be reported in the text. The difference in p-EGFR signal observed in WiDr and CAR1 upon Pevonedistat treatment is probably linked to the presence /absence of autocrine growth factors active on EGFR. Please, quantify at least the mRNA -better the real protein- for the ligands: TGFα, EGF, HB-EGF, AREG, BTC, EPI, EPGN. In my experience, many CRC cell lines release abundant AREG in the medium.

  • The authors could also use present data to complete the observations of their previous paper (https://doi.org/10.1093/jnci/djw209). As they found that Pevonedistat sensitivity was linked to the few cancers (13%) with low EGFR pathway activity, this strong limit could be linked to p-EGFR signaling stabilization by the drug in the cell lines with an active wt EGFR signaling. In these cells, the benefit of p-EGFR stabilization could overwhelm the therapeutic efficacy of NEDD8 inhibition. I suggest to test a CRC cell line with a wt EGFR pathway, and low/null response to Pevonedistat (i.e. Caco2, that produces autocrine AREG), and show the efficacy of a coupled Pevonedistat-EGFR targeting. A true strong synergy between Pevonedistat and EGFR signaling inhibitors could be observed in this condition.

  • Erk1-2 activation is not only triggered by TYR-K receptors, but by numerous stress signals and GPCR: looking at the blots presented in this study (not that of HCA7, that used 48h treated cells, where medium nutrients can be exhausted), it appears that Pevonedistat could directly induce Erk1-2 signaling. The authors should test this hypothesis with a short-term treatment (15’) of cells with Pevonedistat, in fresh serum-free medium (after 24h serum starvation to eliminate most paracrine EGFR activation and lower basal signals).

  • Fig5 a,c,d: add normalized quantifications of signals (phospho/total/vinculin) for EGFR, Akt, Erk1-2

  • Fig5 b,e: these images are of poor interest and do not represent the tumors: sections with very different histological appearance are shown even for the same treatments, moreover the different dominant background colors of tissue slides and variable hematoxylin intensity confuse the reader. Please use serial overlapping IHC sections or multiplex staining of the same slide to analize the staining of the same area with all antibodies and provide an image analysis quantification of markers in this setting for numerous microscopic fields. The number of positive signals must be normalized against the number of cell nuclei in the section, as the multidrug treatment causes the formation of large vacuoles possibly lowering the number of total cells / analyzed surface.

Author Response

We thank the Reviewer for the evaluation. Please, see the attachment of our reply.

Reviewer 3 Report

I have reviewed this manuscript and thought that there are no revisions to publish this one. So, I think this manuscript has sufficient impact for publication in Cancers.

Author Response

(The authors gave the same response as above.)

Round 2

Reviewer 2 Report

Thank you for your revision.